# Oleiferasaponin A_2_, a Novel Saponin from *Camellia oleifera* Abel. Seeds, Inhibits Lipid Accumulation of HepG2 Cells Through Regulating Fatty Acid Metabolism

**DOI:** 10.3390/molecules23123296

**Published:** 2018-12-12

**Authors:** Tai-Mei Di, Shao-Lan Yang, Feng-Yu Du, Lei Zhao, Xiao-Han Li, Tao Xia, Xin-Fu Zhang

**Affiliations:** 1College of Horticulture, Qingdao Agricultural University, Qingdao 266109, China; dtmtea@163.com (T.-M.D.); shaolanyang@126.com (S.-L.Y.); zhaolei_tea@163.com (L.Z.); m17854231961@163.com (X.-H.L.); 2College of Chemistry and Pharmacy, Qingdao Agricultural University, Qingdao 266109, China; fooddfy@126.com; 3State Key Laboratory of Tea Plant Biology and Utilization, Anhui Agricultural University, Hefei 230036, China; xiatao62@126.com

**Keywords:** Camellia oleifera, triterpenoid saponin, oleiferasaponin A_2_, hypolipidemic activity, fatty acid metabolism

## Abstract

A new triterpenoid saponin, named oleiferasaponin A_2_, was isolated and identified from *Camellia oleifera* defatted seeds. Oleiferasaponin A_2_ exhibited anti-hyperlipidemic activity on HepG2 cell lines. Further study of the hypolipidemic mechanism showed that oleiferasaponin A_2_ inhibited fatty acid synthesis by significantly down-regulating the expression of *SREBP-1c, FAS* and FAS protein, while dramatically promoting fatty acid β-oxidation by up-regulating the expression of *ACOX-1, CPT-1* and ACOX-1 protein. Our results demonstrate that the oleiferasaponin A_2_ possesses potential medicinal value for hyperlipidemia treatment.

## 1. Introduction

Hyperlipidemia is a common disease resulting from abnormal lipid metabolism, considered as one of the high-risk factors of inducing cardio-cerebrovascular disease. A large proportion of deaths is caused by cardiovascular disease around the world, more than twice as many deaths as cancer [1], which bring about heavy global medical burden and economic burden. The existing drugs used for lowering lipids are still with non-negligible side-effects and the current status of drug development is less than satisfactory [2,3]. There is an urgent need for safe and efficient anti-hyperlipidemic drugs. Plant extracts have been studied as therapeutic agents for hyperlipidemia [4,5,6]. Saponin is a vital class of natural products, widely used in daily life and production [7,8,9]. Twelve novel triterpenoid saponins have been extracted from *Camellia oleifera* Abel. seeds [10,11,12,13,14,15,16,17,18]. *Camellia oleifera* Abel. is widely cultivated throughout southern China as a commercial crop with abundant edible oil in its seeds. Almost 730,000 tons of defatted tea (*Camellia oleifera*) seeds with 10% saponin are yielded as residue in China annually [17,19], which is a big natural resource. The antimicrobial activity [10,20,21], antiallergic activity [22], antioxidant activity [23] and cytotoxic activity [14,15,24] of saponins have been reported. The research related to the hypolipidemic activity of saponin is scarce [25,26,27], and most hypolipidemic activity research is based on crude saponins [28,29].

Liver is an important organ for lipid metabolism. HepG2 cells not only produce lipids, but also breeds quickly, is easy to cultivate and has high stability. HepG2 cells possess similar cell characteristics to normal human liver cell. HepG2 cell is the representative cell model for studying lipid metabolism, widely used in lipid-regulating drug screening and mechanism research [30,31,32]. Fatty acid synthesis genes, e.g., *SREBP-1c* (sterol-regulatory element-binding protein-1c) [33,34], *FAS* (fatty acid synthase) [35], and *ACC* (acetyl-coenzyme A carboxylase) [36], and fatty acid oxidation genes, e.g., *PPARα* (peroxisone proliferators-activated receptor alpha) [37], *ACOX-1* (acetyl coenzyme A oxidase-1) [38], and *CPT-1* (carnitine palmitoyl transferase-1) [39], are often studied as key genes related to fatty acid metabolism.

In our recent study, we isolated a newly-discovered saponin, oleiferasaponin A_2_, from *Camellia oleifera* Abel. defatted seeds. The structure of oleiferasaponin A_2_ was identified by IR (Infrared Spectroscopy), HR-ESI-MS (High Resolution Electrospray Ionization Mass Spectrometry), NMR (Nuclear Magnetic Resonance) and GC-MS (Gas Chromatography-Mass Spectrometer). The anti-hyperlipidemic activity and mechanism study of oleiferasaponin A_2_ was investigated.

## 2. Results and Discussion

### 2.1. Isolation and Characterization of the Oleiferasaponin A_2_

The structure of oleiferasaponin A_2_ was determined mainly by IR, NMR, HR-ESI-MS and GC-MS (Figure 1A, Table 1, and Appendix A). The molecular formula of oleiferasaponin A_2_ is C_63_H_99_O_28_ deduced from the HR-ESI-MS [M-H]-ion at *m*/*z* 1302.6231 (Appendix A). The IR spectrum (cm^−1^) suggested absorption bands at 3378, 1716, 1618, 1088, and 1045 due to hydroxyl, carbonyl, olefinic, and ether functional groups, respectively (Appendix A). The NMR spectroscopic data of oleiferasaponin A_2_ is shown in Table 1 and Appendix A). The presence of signals at C-21 (δC 78.3, δH 5.94, d, 10.1) and C-22 (δC 73, δH 5.55, d, 10.1) indicate the existence of 21-angeloyl group and 22-isovaleric group, respectively (Table 1). The HMBC (1H detected heteronuclear Multiple Bond Correlation) spectrum (Appendix A) showed correlation between δH 5.94 (H-21) and δC 167.7 (Ang-C-1), δH 5.55 (H-22) and δC 177.4 (IsoA-C-1) (Figure 1B), which confirmed the presence of the Ang group linked to C-21 and Iso group linked to C-22. The NOESY (Nuclear Overhauser Effect Spectroscopy) spectrum (Appendix A) showed the cross peaks between H-22 at δH 5.55 and H-30 at δH 1.10, as well as those between H-16 at δH 4.00 and H-28 at δH 2.95, suggesting that H-22 and H-16 are both β-oriented, that is, iso groups at C-22 and 16-OH group are both α-orientations. The H-3 at δH 3.89 correlated with H-23 at δH 9.48 and H-21 at δH 5.94 correlated with H-29 at δH 0.88, indicating that the glycosidic chain group at C-3 and Ang group at C-21 are β-configured.

Acid hydrolysis and GC-MS analysis (Appendix A) of oleiferasaponin A_2_ revealed there are one unit each of d-glucuronic acid (GlcA), D-glucose (Glu), D-galactose (Gal) and l-arabinose (Ara). The HMBC experiment confirmed the position of the sugar components by the long-ranged correlations between the H-1′ of glucuronic acid and the C-3 of the aglycone, between the H-1′′ of galactose and C-2′ of glucuronic acid, between the H-1‴ of arabinose and C-3′ glucuronic acid, and between the H-1‴′ of glucose and C-2‴ of arabinose (Figure 1B). Based on the above spectroscopy analysis, the structure of novel oleiferasaponin A2 was elucidated to be 16α-hydroxy-21β-*O*-angeloyl-22α-*O*-isovaleric-28-dihydroxymethyleneolean-12-ene-3-*O*-[β-d-galactopyranosyl(1→2)]-[β-d-glucopyranosyl(1→2)-α-L-arabinopyranosyl(1→3)]-β-d-gluco-pyranosiduronic acid. Oleiferasaponin A_2_ with specific structure is a novel compound. The structure of oleiferasaponin A_2_ is composed of two parts, aglycone and sugar moieties, which is similar to oleiferasaponin A_1_, camelliasaponin B_1_ and camelliasaponin B_2_, except for C-21 angeloyl group and C-22 ovaleric group [13,40].

### 2.2. Hypolipidemic Activity of the Oleiferasaponin A_2_

#### 2.2.1. Oleiferasaponin A_2_ Exhibited Hypolipidemic Activity on HepG2 Cell Lines

Oleiferasaponin A_2_ with concentration of 10 μM and 20 μM did not exhibit effective cytotoxic activity on HepG2 cell lines in proliferation bioassay (SRB) (Figure 2A). There are some reports indicating that the cinnamoyl group at C-22 and the free hydroxy group at C-28 play important roles in the anti-proliferative activity of oleanane-type saponins [14,17,41,42]. The cytotoxicity of saponins depends on the key group, as well as the combination of properties at both the aglycone and the sugar moieties. Comparing with amelliasaponin B_1_, which has been reported with obvious cytotoxicity, it seems that the angeloyl group at C-22, rather than C-21, may increase the cytotoxicity [14]. The C-21 angeloyl group and C-22 ovaleric group, as well as the combination of aglycone and sugar moieties make oleiferasaponin A_2_ without cytotoxic activity.

The results of preliminary screening of hypolipidemic activity demonstrated that 10 μM oleiferasaponin A_2_ (*p* = 0.027) and 10 μM simvastatin (*p* = 0.000) exhibited significant hypolipidemic activity on HepG2 cell lines (Figure 2B). The content of triglyceride was determined by detection kit (Figure 2C). Comparing with model group, 10 μM simvastatin (*p* = 0.000) and 10 μM (*p* = 0.012) oleiferasaponin A_2_ significantly inhibited the accumulation of triglyceride. Comparing the structure of oleiferasaponin A_2_ with Chakasaponin I–III [43], Floratheasaponins A–C and theasaponin E1 and E2 [27], we speculated the acyl groups at C21 and C22 may increase the anti-hyperlipidemic activity of oleanane-type saponins, especially, the angeloyl group at C-21.

#### 2.2.2. Oleiferasaponin A_2_ Affected the Expression of Genes Related to Fatty Acid Metabolism

Fluorescent quantitative PCR assay made clear the regulation mechanism of oleiferasaponin A_2_ on the genes related to fatty acid metabolism (Figure 3). Oleiferasaponin A_2_ significantly inhibited the expression of fatty acid synthesis genes, including *SREBP-1c* (*p* = 0.000), *FAS* (*p* = 0.000) and greatly promoted the expression of fatty acid oxidation genes, including *ACOX-1* (*p* = 000) and *CPT-1* (*p* = 0.000). The relative expression of *PPARα* was promoted by oleiferasaponin A_2_ (*p* = 0.208), but without significant variations. Oleiferasaponin A_2_ can reduce lipid accumulation by inhibiting fatty acid synthesis and promoting fatty acid β-oxidation, which can explain the decrease of triglyceride content (Figure 2C). The expression of *SREBP-1c* (*p* = 0.000) was significantly inhibited by simvastatin. The relative expression of fatty acid synthesis genes, *FAS* (*p* = 0.005) and *ACC* (*p* = 0.000), and fatty acid β-oxidation genes, *PPARα* (*p* = 0.008)*, ACOX-1* (*p* = 0.000) and *CPT-1* (*p* = 0.000), were significantly elevated by the simvastatin, which revealed the action mechanism of simvastatin. Simvastatin realized lipid lowering by inhibited *SREBP-1c* and promoting fatty acid β-oxidation. Comparing with simvastatin, oleiferasaponin A_2_ showed excellent inhibition action on the expression of *FAS*, which plays a key role in fatty acid synthesis.

#### 2.2.3. Oleiferasaponin A_2_ Affected the Expression of Proteins Related to Fatty Acid Metabolism

Western blotting results showed the effect of oleiferasaponin A_2_ and simvastatin on the proteins related to fatty acid metabolism (Figure 4). The expression of ACOX-1 was significantly promoted by 10 μM oleiferasaponin A_2_, which was superior to the promotion effect of 10 μM simvastatin. The expression of FAS was significantly inhibited by 10 μM oleiferasaponin A_2_, while promoted by 10 μM simvastatin. FAS is responsible for the formation of free fatty acids, such as palmitate from acetyl-CoA and malonyl-CoA [44]. ACOX-1 is the rate-limiting enzyme of the first dehydrogenation reaction in peroxysomal lipase oxygenation [45]. Therefore, FAS and ACOX-1 play an important role in fat deposition. Oleiferasaponin A_2_ performs lipid-lowering effect by reducing the fatty acid production and accelerating fatty acid oxidation.

## 3. Materials and Methods

### 3.1. General

The following equipment was used: 3000-Da nanofiltration membrane (SJM, Hefei, China), AB-8 macroporous resin column (Bonc, Cangzhou, China), ordinary-phase silica gel column (200–300 mesh, Anhui Liangchen Silicon Material Co. Ltd., Huoshan, China) and a Varian Prostar HPLC instrument (Model 325) (Varian, Mulgrave, Australia). The HPLC purifications were run on Agilent 1260 HPLC (Agilent, Palo Alto, CA, USA). IR (infrared) spectra were recorded on Nicolet iN10 (Thermo Scientific Instrument Co., Boston, MA, USA) with KBr pellets. NMR spectra were measured on an AVANCE III (600 MHz) spectrometer (Bruker, Fallanden, Switzerland) using methanol-d4 as solvent g methanol-d4 (Sigma-Aldrich, St. Louis, MO, USA). HR-ESI-MS were determined on an Electrostatic Field Orbital Trap Mass Spectrometer (Thermo Scientific, Bremen, Germany) using an ESI source.

### 3.2. Plant Material

The *Camellia oleifera* Abel. defatted seeds were obtained from a manufacture factory in Shucheng, Anhui province, China, in October 2013. The plant material was identified by one of the authors (X.F.Z.), and was deposited in State Key Laboratory of Tea Plant Biology and Utilization, Anhui Agricultural University.

### 3.3. Extraction and Isolation

The defatted seed powder (10 kg) was extracted three times with methanol at 60 °C under reflux for 3 h each time. The extract solvent was evaporated under reduced pressure. Next, the concentrated solution (1.3 kg) was suspended in water and successively subjected to nanofiltration membrane, AB-8 macroporous resin column (Bonc, Cangzhou, Herbei, China) (chromatography with stepwise gradients of water and ethanol (100:0, 70:30, 30:70, and 0:100, *v*/*v*)), and ordinary-phase silica gel column [CHCl_2_:CH_3_OH:H_2_O (80:60:5, *v*/*v*)] to yield a high purity fraction (1.6 g). Further isolation and purification were performed by HPLC [MeOH: H_2_O (30:70)] to yield two purity fractions [Fr. 1 (0.07 g), Fr. 2 (0.21 g)], which were purified by HPLC [acetonitrile-0.2% AcOH: H_2_O (41:59, *v*/*v*)]. Then, the oleiferasaponin A_2_ (12.9 mg) was obtained from the second fraction. All sample extraction and isolation methods were according to Zhang et al. [13].

### 3.4. Acid Hydrolysis and GC-MS Analysis

The method of acid hydrolysis and GC-MS analysis was according to Zong et al. [14]. Oleiferasaponin A_2_ was dissolved in 1 M HCI (1 mL) for 3 h at 90 °C and extracted with chloroform, and then evaporated under N_2_ flow. The residue was dissolved in 0.2 mL pyridine containing L-cysteine methyl ester hydrochloride (10 mg/mL) and reacted at 70 °C for 1 h, and then evaporated under N_2_ flow again. After being concentrated, 0.2 mL trimethylsilylimidazole was added for derivatization reaction, and reacted at 70 °C for another 1 h. The reaction mixture was partitioned between n-hexane and water. The organic phase was analyzed by GC-MS (Agilent, Palo Alto, CA, USA) (injector temperature was 280 °C; the initial oven temperature was 160 °C for 1 min, linearly increased to 200 °C at 6 °C/min, further linearly increased to 280 °C at 3 °C/min and held for 5 min). The standard sugar samples were subjected to the same reaction and GC-MS conditions.

### 3.5. Cell Viability Assay

Human liver tumor cell (HepG2) lines were obtained from Qingdao Marine Biomedical Research Institute Limited by Share Ltd. Testing Center (Qingdao, China). Cells were cultured in DMEM complete medium supplemented with 10% fetal bovine serum, 2 mM l-glutamine, 100 U mL^−1^ penicillin and 100 μg mL^−1^ streptomycin at 37 °C in a 5% CO_2_ humidified atmosphere. The culture medium was refreshed every other day. After 80% of the cells were fused, cells were kept in logarithmic phase by trypsinization and subculturing [14,16].

Human tumor cell lines in logarithmic phase were seeded in 96-well plates at 4 × 10^3^ cells per well (180 μL per well), and incubated for 24 h. After 24 h, an additional complete medium with no additions (negative control), 0.1% DMSO (solvent control), 1 μM adriamycin (positive control), or 20 μM or 10 μM test saponins (sample control) was incubated for 72 h. Then, 50% (*m*/*v*) ice-cold trichloroacetic acid was added to the medium to fix cells. After staining by Sulforhodamine B, tris solution (150 μL per well) was added to culture medium. Absorbance values were measured at 540 nm using an enzyme-linked immunosorbent Reader. The inhibition rate of cell proliferation was calculated as: Inhibition rate (%) = [(OD_540_ (Negative control) − OD_540_ (Sample control))/OD_540_ (Negative control)] × 100% [14].

### 3.6. Preliminary Screening of Hypolipidemic Activity

HepG2 cells (1.2 × 10^3^ per well) at logarithmic phase were seeded in 96-well plates (100 μL per well). After 70–80% fused, the medium was replaced with free-blood serum DMEM (80 μL per well), and starved for 12 h. Blank group with 20 μL free-blood serum DMEM, model group, SVST group and A_2_ group were added with 80 μM Oleic Acid (10 μL per well). Then, model group was supplemented with 10 μL serum-free medium, and 10 μM simvastatin and oleiferasaponin A_2_ were added to positive control and treatment group, respectively, incubated for 24 h. After incubating for 24 h, the medium was removed. Cells were washed with PBS once, fixed with 4% paraformaldehyde (80 μL per well) for 1 h, washed with PBS again and rinsed with 60% isopropanol for 10 min. Then, cells were stained with 60 μL 0.3% Oil Red O solution (Sigma O0625) for 1 h, washed with PBS for thrice, dissolved by DMSO (100 μL per well). The OD value (358 nm) was measured by ELIASA (SpectraMax i3) (Molecular devices, Shanghai, China).

### 3.7. Triglyceride Test

HepG2 cells (2.5 × 10^5^ per well) at logarithmic phase were seeded in 6-well plates (2 mL per well). After 12 h, the medium was replaced with free-blood serum DMEM (2 mL per well), and starved for 12 h. Blank group was supplemented with 1 mL free-blood serum DMEM, and the other groups were added with 80 μM Oleic Acid (300 μL per well). Then, model group was supplemented with 700 μL serum-free medium, and 300 μL simvastatin and oleiferasaponin A_2_ (10 μM) were added to positive control and treatment group, respectively, and incubated for 24 h. After incubating 24 h, the medium was removed. Cells were washed with PBS twice. The content of triglyceride was tested according to K622-100 triglyceride test kit (BioVision, San Francisco, CA, USA) instructions.

### 3.8. Fluorescent Quantitative PCR

After incubating 24 h, for the medium was removed. Cells were washed with PBS twice. Trizol (500 μL per well) was added for cell lysis, then homogenized and placed at room temperature for 5 min. One hundred microliters of chloroform were added, then shook for 15 s, let stand for 3 min, and centrifuged at 12,000 r/min, 4 °C for 15 min. The supernatant was transferred to centrifuge tube, 250 μL of isopropanol were added, shook, let stand, and then centrifuged. The supernatant was removed, and the remaining was resuspended with 75% ethyl alcohol and centrifuged. Supernatant was removed and aired. Twenty microliters of DEPC were added for RNA hydrolysis, and the concentration determined. cDNA was obtained through reverse transcription reaction, and then the gene relative expression was tested by real-time fluorescent quantitative PCR (LightCycler 96, Roche Life Science, Basel, Switzerland) under following reaction systems: 12.5 μL SYBR Premix Ex Taq ll (Tli RNaseH Plus) (2×), 1.0 μL PCR Forward Primer (10 μM), 1.0 μL PCR Reverse Primer (10 μM), 2 μL cDNA solution, and 8.5 μL dH_2_O. Primer set used for q-PCR analysis was designed by Primer 3 (Appendix A) (http:// Frodo.wi.mit.edu/cgi-bin/primer3-www.cgi).

### 3.9. Western Blot Analysis

After incubating for 24 h, the medium was removed. Cells were washed with PBS twice. Two hundred microliters of RIPA (containing 10 μL PMSF) were added for cell lysis. After being homogenized on ice for 30 min, the samples were centrifuged at 12,000 r/min for 5 min, and the supernatants were collected for Western blot analysis. The concentration of protein was measured by BCA kit (Thermo Fisher scientific, Waltham, MA, USA). The lysate containing SDS-PAGE protein loading buffer was placed in boiling bath for protein albumen metamorphism, centrifuged at 12,000 rpm for 2 min and the supernatant was kept at −80 °C for SDS-PAGE. Equal amounts of protein were subjected to SDS-PAGE (12% separating gel and 5% stacking gel), and then transferred to PVDF membranes, which was placed in sealing solution and blocked for 2 h at room temperature. The PVDF membranes was incubated and shocked overnight with primary antibodies (FAS:1:200; ACOX-1:1:1000), and washed with TBST for 3 times. After washing with TBST, the PVDF membranes were incubated with secondary antibodies (1:2000) for 2 h at room temperature and washed with TBST thrice. PVDF membranes with ECL reagent (regent A:regent B is 1:1) were incubated for 30 s, and then chemiluminescent signals were visualized and analyzed using scanned gel imaging systems (GIS-2008) (Analytik Jena AG, Jena, Germeny). The method of Western blot refers to Zong et al. [16].

## 4. Conclusions

Our present study reveals that oleiferasaponin A_2_, a novel compound from *Camellia oleifera* defatted seeds, can accomplish lipid-lowering of HepG2 cells. Oleiferasaponin A_2_ can down-regulate *SREBP-1c* and *FAS* expression and FAS protein expression, and up-regulate *ACOX-1*, *CPT-1*, and *PPARα* expression and ACOX-1 protein expression, which are related to fatty acid metabolism. Thus, we have primarily testified that oleiferasaponin A_2_ performs anti-hyperlipidemic effect by repressing fatty acid synthesis and accelerating fatty acid oxidation in vitro. The structure identification and anti-hyperlipidemic activity study of oleiferasaponin A_2_ will promote understanding of the structure–activity relationship of oleanane-type saponins and expedite drug development, while improving the utilization ratio of natural resources. In the future, animal experiments of hypolipidemic activity should be carried out, and the medicinal value of oleiferasaponin A_2_ should be excavated in more detail.

## Figures and Tables

**Figure 1 molecules-23-03296-f001:**
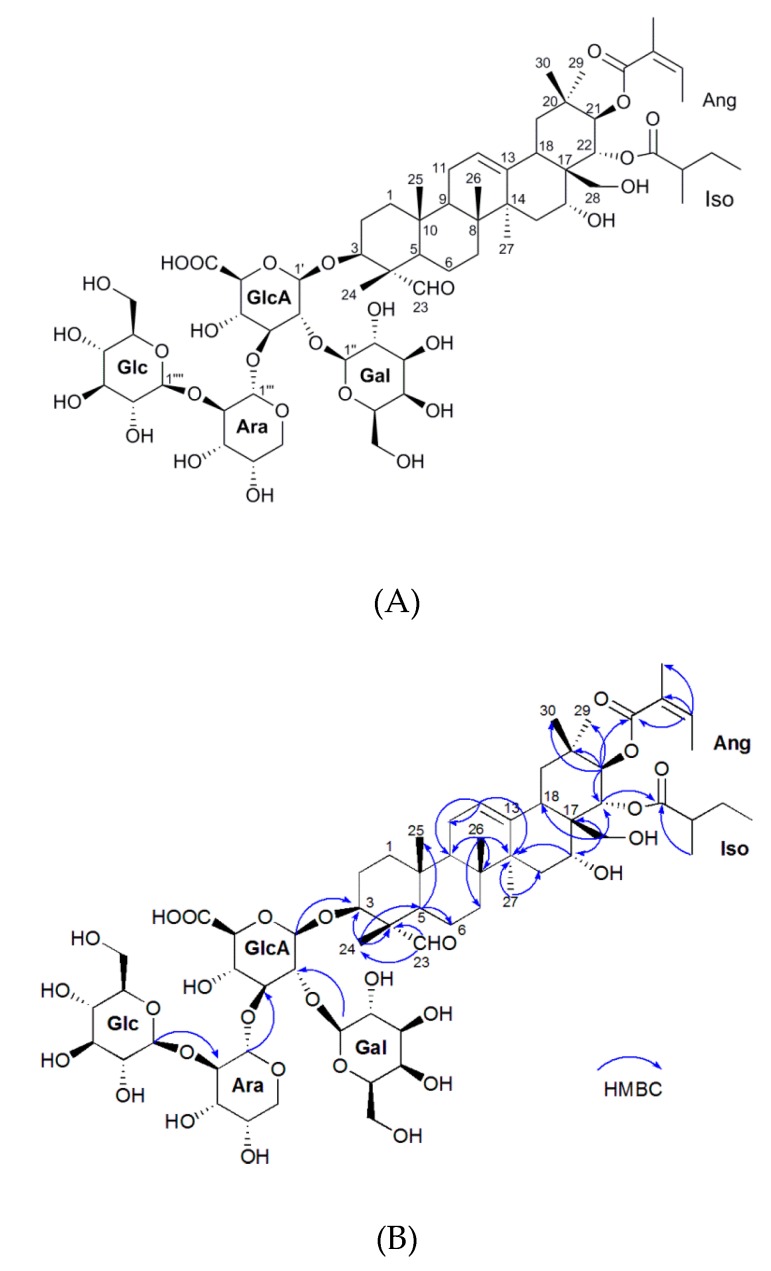
(**A**) Structure of oleiferasaponin A_2_; and (**B**) Key HMBC correlations of oleiferasaponin A_2_.

**Figure 2 molecules-23-03296-f002:**
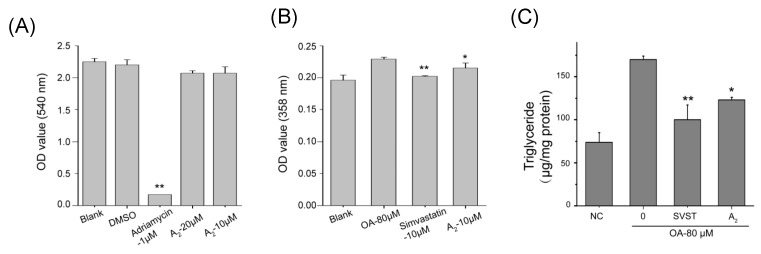
(**A**) The effect of oleiferasaponin A_2_ on HepG2 cells proliferation; (**B**) the lipid lowering activity of oleiferasaponin A_2_ under oleic acid treatment; and (**C**) the content of triglyceride under OA inducement. NC, blank control without oleic acid; 0, model group with 80 μM oleic acid; SVST, group with 80 μM oleic acid and 10 μM simvastain; A_2_, group with 80 μM oleic acid and 10 μM oleiferasaponin A_2_. ** *p* < 0.01, * *p* < 0.05.

**Figure 3 molecules-23-03296-f003:**
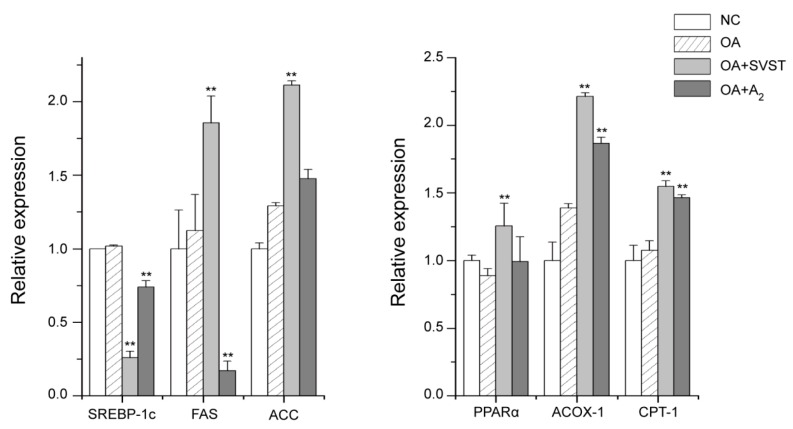
The relative expression of genes related to fatty acid metabolism on fatty acid synthesis genes (*SREBP-1c, FAS* and *ACC*) and fatty acid *β*-oxidation genes (*PPARα, ACOX-1* and *CPT-1*). NC, blank control; OA, 80 μM oleic Acid; OA + SVST, 80 μM Oleic Acid + 10 μM simvastatin; OA + A_2_, 80 μM Oleic Acid + 10 μM Oleiferasapoin A_2_. Compared to the control, ** *p* < 0.01.

**Figure 4 molecules-23-03296-f004:**
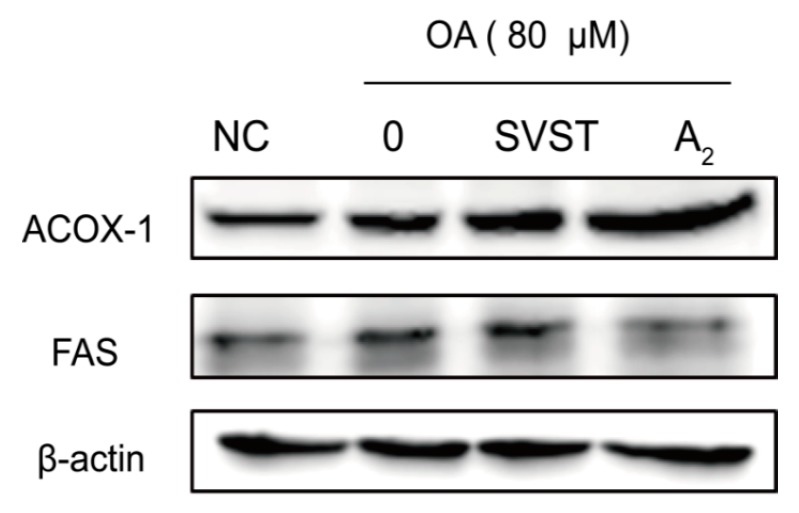
The relative expression of proteins related to fatty acid metabolism. NC, blank control without oleic acid; 0, model group with 80 μM oleic acid; SVST, group with 80 μM oleic acid and 10 μM simvastain; A_2_, group with 80 μM oleic acid and 10 μM oleiferasapoin A_2_; ACOX-1, fatty acid oxidation protein; FAS, fatty acid synthesis protein; β-actin, reference protein.

**Table 1 molecules-23-03296-t001:** NMR spectroscopic data for oleiferasaponin A_2_ (in methanol-*d*_4_).

Position	*δ* _C_	*δ* _H_	Position	*δ* _C_	*δ* _H_
1	38.0	1.14 m, 1.73 m	21-O-Ang		
2	24.3	1.81 m, 2.05 m	Ang-1	167.7	
3	84.9	3.89 m	Ang-2	127.7	
4	55.0		Ang-3	139.0	6.18 m
5	47.5	1.38 m	Ang-4	14.8	1.94 m
6	19.8	0.96 m	Ang-5	19.7	1.87 s
7	31.8	1.29 m, 1.64 m	22-O-Iso		
8	39.9		Iso-1	177.4	
9	46.5	1.81 m	Iso-2	41.3	2.33 m
10	35.6		Iso-3	26.3	1.43 m, 1.65 m
11	23.3	1.98 m	Iso-4	10.8	0.89 m
12	123.6	5.41, t, 3.7	Iso-5	15.6	1.05, d, 6.8
13	141.6		GlcA-1′	103.4	4.42, d, 7.7
14	41.0		GlcA-2′	76.8	3.77 m
15	33.4	1.37 m, 1.70 m	GlcA-3′	81.7	3.88 overlap
16	68.1	4.00 m	GlcA-4′	69.1	3.84 overlap
17	47.2		GlcA-5′	75.5	3.56 m
18	39.4	2.65 m	GlcA-6′	177.4	
19	46.3	1.21 m, 2.69 m	Gal-1”	101.3	5.04, d, 7.3
20	35.6		Gal-2”	72.1	3.50 m
21	78.3	5.94, d, 10.1	Gal-3”	73.4	3.76 m
22	73.0	5.55, d, 10.1	Gal-4”	69.1	3.84 m
23	209.4	9.48 s	Gal-5”	74.8	3.30 m
24	9.5	1.19 s	Gal-6”	61.1	3.69 m
25	15.1	2.00 s	Ara-1‴	100.2	5.03, d, 7.3
26	15.9	0.96 s	Ara-2‴	82.4	3.68 m
27	26.4	1.53 s	Ara-3‴	69.6	3.53 m
28	63.0	2.95 m, 3.77 m	Ara-4‴	68.1	4.00 m
29	28.3	0.88 s	Ara-5‴	65.9	3.21 m, 3.97 m
30	19.0	1.10 s	Glc-1‴′	106.2	4.53, d, 7.7
			Glc-2‴′	75.1	3.65 m
			Glc-3‴′	73.6	3.51 overlap
			Glc-4‴′	69.6	3.53 overlap
			Glc-5‴′	76.4	3.36 m
			Glc-6‴′	61.2	3.80 m

^1^H (*δ* ppm; *J* in Hz; s, Single peak; d, Double peaks; m, multiple peaks) and ^13^C-NMR (*δ* ppm).

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
