# Peer review of "Oleiferasaponin A2, a Novel Saponin from Camellia oleifera Abel. Seeds, Inhibits Lipid Accumulation of HepG2 Cells Through Regulating Fatty Acid Metabolism"

_molecules, 2018, doi:10.3390/molecules23123296_

Round 1

Reviewer 1 Report

The manuscript „Oleiferasaponin A2, a novel saponin from Camellia oleifera Abel. seeds inhibits lipid accumulation of HepG2 Cells through regulating fatty acid metabolism“ of Tai-mei Di and colleagues, reported on a new triterpenoid saponin that was isolated from Camellia oleifera seeds. The new saponin exhibited anti-hyperlipidemic activity on HepG2 cell lines. The Taime Di and coworkers further characterized the saponin in detail. Deatailed results on a potential medicinal value for hyperlipidemia treatment is discussed.

The data are interesting and the manuscript seems tob e carefully conducted. However, this reviewer misses a meaningful discussion oft he importance of the results and information on the current state oft he scientic research within this field.

There are also some spelling, typing and style errors that should be carefully revised in the next version oft he manuscript.

Line 34: Abel.? Explanation?

Line 34: seed instead of Seed

Figure 2, Figure 3: It would be helpful, if the authors could include the real p-values oft he individual dataset in the figure.

Whole manuscript: A space between the number and the unit is missing.

Figure 4 legend: Need a more detailed description.

Table 1 should be incorporated in the supplemental material part.

Author Response

Dear reviewer,

 We are very grateful to your comments for our manuscript titled: “Oleiferasaponin A2, a novel saponin from Camellia oleifera Abel. seeds inhibits lipid accumulation of HepG2 Cells through regulating fatty acid metabolism”, and we are appreciated the efforts that you have made for our manuscript in review process. Your precious advices are very professional, accurate and helpful for us. We have amended the relevant part in manuscript according to your comments and suggestions. At the same time, we will pay more attentions to the aspects that you have mentioned in the future study. We have addressed point-by-point the details of the revisions in the manuscript and my responses to the comments. Changes in the revised manuscript are marked in highlighted font.

Response to Reviewer 1 Comments

Point 1: A meaningful discussion of the importance of the results and information on the current state of the scientic research within this field.

Response 1:

The discussion of importance of the results was added in Line 314 –line 317 Page 9.

The structure identification and anti-hyperlipidemic activity study of oleiferasaponin A2 will promote understanding the structure-activity relationship of oleanane-type saponins and expedite drug development, meanwhile, improve the utilization ratio of natural resource.

The discussion of information on the current state of the scientic research within this field was added in Line 116 –line123 page 4, line 128 –line131 Page 5.  

There are some reports indicating that the cinnamoyl group at C-22 and the free hydroxy group at C-28 play important roles in the anti-proliferative activity of oleanane-type saponins [14,17,41,42]. The cytotoxicity of saponins depends on the key group, as well as the combination of properties at both the aglycone and the sugar moieties. Comparing with amelliasaponin B1 which has been reported with obvious cytotoxicity, it seems that the angeloyl group at C-22, rather than C-21 may increase the cytotoxicity [14]. The C-21 angeloyl group and C-22 ovaleric group, as well as the combination of aglycone and sugar moieties make oleiferasaponins A2 without cytotoxic activity. Comparing  the structure of oleiferasaponin A2 with Chakasaponin I–III [43], Floratheasaponins A–C and theasaponin E1 and E2[44], we speculated the acyl groups at C21 and C22 may increase the anti-hyperlipidemic activity of oleanane-type saponins, especially, the angeloyl group at C-21.

Point 2:  There are also some spelling, typing and style errors that should be carefully revised in the next version of the manuscript.

Response 2: The spelling, typing and style errors have been carefully revised in the manuscript.

Point 3: Line 34: Abel.? explanation? Line 34: seed instead of Seed

Response 3: Abel. : Clarke Abel (5 September 1780 – 14 November 1826) was a British surgeon and naturalist. seed instead of Seed. Line 34 Page 1.

Point 4:  Figure 2, Figure 3: It would be helpful, if the authors could include the real p-values of the individual dataset in the figure.

Response 4: The real p-values of the individual dataset in the figure have been added in the article. (Highlighted font)

Point 5: Whole manuscript: A space between the number and the unit is missing.

Response 5: A space between the number and the unit was added.

Point 6: Figure 4 legend: Need a more detailed description.

Response 6: More detailed description of figure 4 was added.

Point 7:  Table 1 should be incorporated in the supplemental material part.

Response 7: Table 1 has been incorporated in the supplemental material part.

We have made improvement according to your professional comments and suggestions. If there are any other questions regarding our manuscript, please do not hesitate to contact us as we really appreciate the opportunity to publish our research in Molecules. We look forward to hearing from you soon.

Sincerely yours,

Xin-Fu Zhang, professor

Qingdao Agricultural University, Qingdao 266109, P.R. China

E-mail: zxftea@163.com

Reviewer 2 Report

structural information was extensive for this compound, and appeared to support the structural assignment. The grammatical mistakes made some parts difficult to read. The Figure 3 did not have concentrations listed - probably the same as Figure 2 perhaps? The results needed a bit more elaboration to make what they found more clear to the reader who might not be as familiar with this work. For example, in Figure 3 FAS column simvastatin and A2 show results that are opposite in nature. Otherwise it is a very simple paper using standard techniques to show activity of an interesting natural product. 

Author Response

Dear reviewer,

 We are very grateful to your comments for our manuscript titled: “Oleiferasaponin A2, a novel saponin from Camellia oleifera Abel. seeds inhibits lipid accumulation of HepG2 Cells through regulating fatty acid metabolism”, and we are appreciated the efforts that you have made for our manuscript in review process. Your precious advices are very professional, accurate and helpful for us. We have amended the relevant part in manuscript according to your comments and suggestions. At the same time, we will pay more attentions to the aspects that you have mentioned in the future study. We have addressed point-by-point the details of the revisions in the manuscript and my responses to the comments. Changes in the revised manuscript are marked in highlighted font.

Response to Reviewer 2 Comments

Response 1:

Point 1:  The grammatical mistakes made some parts difficult to read.

Response 1:  Grammatical mistakes have been carefully revised in the manuscript.

Point 2:  The Figure 3 did not have concentrations listed - probably the same as Figure 2 perhaps?

Response 2: More detailed description of figure 3 legend was added. Line 178 –line181 Page 6.

Point 3: The results needed a bit more elaboration to make what they found more clear to the reader who might not be as familiar with this work.

Response 3: More elaboration has been added in the manuscript. Line 116–line123 Page 4 line 128- line131 Page 5, Line 184–line190 Page 5.

There are some reports indicating that the cinnamoyl group at C-22 and the free hydroxy group at C-28 play important roles in the anti-proliferative activity of oleanane-type saponins [14,17,41,42]. The cytotoxicity of saponins depends on the key group, as well as the combination of properties at both the aglycone and the sugar moieties. Comparing with amelliasaponin B1 which has been reported with obvious cytotoxicity, it seems that the angeloyl group at C-22, rather than C-21 may increase the cytotoxicity [14]. The C-21 angeloyl group and C-22 ovaleric group, as well as the combination of aglycone and sugar moieties make oleiferasaponins A2 without cytotoxic activity.

Comparing  the structure of oleiferasaponin A2 with Chakasaponin I–III [43], Floratheasaponins A–C and theasaponin E1 and E2[44], we speculated the acyl groups at C21 and C22 may increase the anti-hyperlipidemic activity of oleanane-type saponins, especially, the angeloyl group at C-21.

The expression of ACOX-1 was significantly promoted by 10 μM oleiferasaponin A2, which was superior to the promotion effect of 10 μM simvastatin. The expression of FAS was significantly inhibited by 10 μM oleiferasaponin A2, while was promoted by 10 μM simvastatin. FAS is responsible for the formation of free fatty acids, like palmitate, from from acetyl-coa and malonyl-CoA [45]. ACOX-1 is the rate-limiting enzyme of the first dehydrogenation reaction in the peroxysomal lipase–oxygenation [46]. Therefore, FAS and ACOX-1 play an important role in fat deposition. Oleiferasaponin A2 performs lipid-lowing effect by reducing the fatty acid production and accelerating fatty acid oxidation.

We have made improvement according to your professional comments and suggestions. If there are any other questions regarding our manuscript, please do not hesitate to contact us as we really appreciate the opportunity to publish our research in Molecules. We look forward to hearing from you soon.

Sincerely yours,

Xin-Fu Zhang, professor

Qingdao Agricultural University, Qingdao 266109, P.R. China

E-mail: zxftea@163.com

Tel: +86 0532-88030231

Fax: +86 0532-88030231

Reviewer 3 Report

The current study was objected to investigate new saponin which was isolated from Camellia oleifera Abel. The study design is good and results seems to be interesting. However, in introduction and discussion, the rationale for studying anti-hyperlipidemic activity and discussion comparing other saponins in terms of activity are limited. These points should be added.

Author Response

Dear reviewer,

 We are very grateful to your comments for our manuscript titled: “Oleiferasaponin A2, a novel saponin from Camellia oleifera Abel. seeds inhibits lipid accumulation of HepG2 Cells through regulating fatty acid metabolism”, and we are appreciated the efforts that you have made for our manuscript in review process. Your precious advices are very professional, accurate and helpful for us. We have amended the relevant part in manuscript according to your comments and suggestions. At the same time, we will pay more attentions to the aspects that you have mentioned in the future study. We have addressed point-by-point the details of the revisions in the manuscript and my responses to the comments. Changes in the revised manuscript are marked in highlighted font.

Response to Reviewer 3 Comments

Point 1: In introduction and discussion, the rationale for studying anti-hyperlipidemic activity and discussion comparing other saponins in terms of activity are limited. These points should be added.

Response 1:

The rationale for studying anti-hyperlipidemic activity has been added. Line 42 –line 50 Page 1-2.

Liver is important organ for lipid metabolism. HepG2 cell not only produce lipid properly, but also is breeding fast, easy cultivation and high stability. HepG2 cell possesses similar cell characteristics with normal human liver cell. HepG2 cell is the representative cell model for studying lipid metabolism, widely used in lipid-regulating drugs screening and mechanism research [30-32]. Fatty acid synthesis genes, SREBP-1c (sterol-regulatory element-binding protein-1c) [33,34], FAS (fatty acid synthase) [35], ACC (acetyl-coenzyme A carboxylase) [36] and fatty acid oxidation genes, PPARα (peroxisone proliferators-activated receptor alpha) [37], ACOX-1 (acetyl coenzyme A oxidase-1) [38], CPT-1 (carnitine palmitoyl transferase-1) [39] were often studied as key genes related to fatty acid metabolism

The discussion regarding comparing other saponins in terms of activity has been added. Line 116 –line123 page 4, line 128 –line131 Page 5.

There are some reports indicating that the cinnamoyl group at C-22 and the free hydroxy group at C-28 play important roles in the anti-proliferative activity of oleanane-type saponins [14,17,41,42]. The cytotoxicity of saponins depends on the key group, as well as the combination of properties at both the aglycone and the sugar moieties. Comparing with amelliasaponin B1 which has been reported with obvious cytotoxicity, it seems that the angeloyl group at C-22, rather than C-21 may increase the cytotoxicity [14]. The C-21 angeloyl group and C-22 ovaleric group, as well as the combination of aglycone and sugar moieties make oleiferasaponins A2 without cytotoxic activity.

Comparing  the structure of oleiferasaponin A2 with Chakasaponin I–III [43], Floratheasaponins A–C and theasaponin E1 and E2[44], we speculated the acyl groups at C21 and C22 may increase the anti-hyperlipidemic activity of oleanane-type saponins, especially, the angeloyl group at C-21.

We have made improvement according to your professional comments and suggestions. If there are any other questions regarding our manuscript, please do not hesitate to contact us as we really appreciate the opportunity to publish our research in Molecules. We look forward to hearing from you soon.

Sincerely yours,

Xin-Fu Zhang, professor

Qingdao Agricultural University, Qingdao 266109, P.R. China

E-mail: zxftea@163.com

Tel: +86 0532-88030231

Fax: +86 0532-88030231

Round 2

Reviewer 1 Report

The quality of the manuscript has been improved by the revisions.

This reviewer feels that the manuscript is now acceptable for publication.